# Numerical Study on the Critical Frequency Response of Jet Engine Rotors for Blade-Off Conditions against Bird Strike

**Saeed Badshah** [1],*[ID]**, Ahsan Naeem** [1]**, Amer Farhan Rafique** [2][ID]**, Ihsan Ul Haq** [3][ID]
**and Suheel Abdullah Malik** [3][ID]

1   Department of Mechanical Engineering, International Islamic University Islamabad, Islamabad 44000,
    Pakistan; mech_ahsan113@yahoo.com
2   Department of Aerospace Engineering, Faculty of Engineering, King Abdulaziz University,
    Jeddah 21589, Saudi Arabia; afrafique@kau.edu.sa
3   Department of Electrical Engineering, International Islamic University Islamabad,
    Islamabad 44000, Pakistan; ihsanulhaq@iiu.edu.pk (I.U.H.); suheel.abdullah@iiu.edu.pk (S.A.M.)
*   Correspondence: saeed.badshah@iiu.edu.pk

**Abstract:** Vibrations are usually induced in aero engines under their normal operating conditions. Therefore, it is necessary to predict the critical frequencies of the rotating components carefully. Blade deformation of a jet engine under its normal operating conditions due to fatigue or bird strike is a realistic possibility. This puts the deformed blade as one of the major safety concerns in commercially operating civil aviation. A bird strike introduces unbalanced forces and non-linearities into the engine rotor system. Such dynamic behavior is a primary cause of catastrophic failures. The introduction of unbalanced forces due to a deformed blade, as a result of a bird strike, can change the critical frequency behavior of engine rotor systems. Therefore, it is necessary to predict their critical frequencies and dynamic behavior carefully. The simplified approach of the one-dimensional and two-dimensional elements can be used to predict critical frequencies and critical mode shapes in many cases, but the use of three-dimensional elements is the best method to achieve the goals of a modal analysis. This research explores the effect of a bird strike on the critical frequencies of an engine rotor. The changes in critical mode shapes and critical frequencies as a result of a bird strike on an engine blade are studied in this research. Commercially available analysis software ANSYS version 18.2 is used in this study. In order to account for the material nonlinearities, a Johnson Cook material model is used for the fan blades and an isotropic–elastic–plastic–hydrodynamic material model is used for modeling the bird. The bird strike event is analyzed using Eularian and smoothed particle hydrodynamics (SPH) techniques. A difference of 0.1% is noted in the results of both techniques. In the modal analysis simulation of the engine rotor before and after the bird strike event, the critical failure modes remain same. However, a change in the critical frequencies of the modes is observed. An increase in the critical frequencies and excitation RPMs (revolution per minute) of each mode are observed. As the mode order is increased, the higher the rise in critical frequency and excitation RPMs. Also, a change in the whirl direction of the different modes is noted.

**Keywords:** critical frequency response; bird strike; blade-off condition; FEA

## 1. Introduction

The study of the critical frequencies of the rotating component has prime significance in the design of many structural components. Traditionally, beam theories are used to model the shaft of the rotating components, where the discs of the rotating components are represented by introducing a point mass or

a concentrated mass with a specific moment of inertia. However, the beam theory approach is based on the fact that deformations in the discs or blades are ignored, and the shaft is also assumed to be a slender body. This may provide accurate results, but in order to address deficiencies, higher order elements should be used [1]. The simplified approach of the one-dimensional and two-dimensional elements can be used in many cases, but the use of three-dimensional elements is the best method to achieve the goals of a dynamic analysis. One of the advantages of using three-dimensional elements is to directly develop a finite element model of the rotating component rather using any other mathematical approximation. Another advantage of using a 3D element is that the gyroscopic effects of a rotating system can also be studied [2].

The blade deformation of a turbo jet or turbo fan under normal operating conditions, due to fatigue or bird strike, is a realistic possibility. This puts the blade-off condition as one of the major safety concerns in commercially operating civil aviation [3].

As per the records of the International Bird Strike Committee, 55 fatal accidents have been reported, 108 damaged aircraft, and 277 lives have been lost due to bird strike events. The bird strike on an engine blade also introduces an unbalanced rotating force that can affect the excitation revolutions of the critical frequencies of an engine rotor [4].

The deformed blade of a jet engine as a result of a bird strike event introduces non-linearities in the critical frequency behavior of the system. The disk also starts to rotate from the center towards the heavier side of the disk, because of the suddenly introduced imbalance in the system due to the blade deformation. It is necessary to predict the behavior of such non-linearities on the critical frequency response of an engine rotor system. One-dimensional and two-dimensional models based on Euler and Timoshenko's beam theories cannot address the kinematic description of an engine rotor in the case of a blade-off condition as result of a bird strike event [1,5]. Therefore, a 3D formulation is required for the complete analysis of the rotating bodies. Gyroscopic effects can also be accounted for using a 3D model [6].

The literature is scarce regarding the use of 3D models for assessing the critical frequencies of jet engines due to blade deformation in the event of a bird strike. Therefore, the finite element approach provides a better solution to simulate such unevenness of forces in the rotor of a jet engine. This research focuses on using commercially available codes, such as ANSYS, for the dynamic analysis of the 3D finite element model of aero engines with a blade-off condition in the event of a bird strike.

## 2. Materials and Methods

### 2.1. Foreign Object Damage on Engine Rotor

The primary fan blades on civil engines are tremendously susceptible to bird attack influence, which might lead to tip removal or damage, aerodynamic performance reductions, an imbalance of the rotor, damage, or even a puncture in the engine casing. Under such conditions, an engine can disturb airplane security as it is its only source of power. Therefore, numerous republics have set forth comparative airworthiness standards regarding bird attack for aero-engines, such as the CCAR-33R2 in China and the FAR (Federal Aviation Regulations) in the United States of America [4].

### 2.2. Use of 3D-Elements in Critical Frequency Analysis

The use of one-dimensional elements, which are based on plate theory or plane stress assumptions, can be found in the literature for determining the critical frequencies of the rotating components. Bauer [7] and Curti et al. [8] have calculated critical speeds of critical frequencies for metallic shafts using one-dimensional elements based on Euler–Bernoulli and Timoshenko's beam theories. Similarly, Bauchau and Chen et al. [9] also used these theories to analyze walled composite shafts.

On the other hand, by combining one-, two-, and three-dimensional methods, several rotor settings were designed by Jang and Lee [10]. Their approach was to combine the FEM with the sub-structure synthesis in the definition of the spindles of the rotating disks in order to find critical frequencies.

The 2D elements, which were based on Kirchoff's theory, were used to model the disk, whereas the stationary components and spindle were modeled using the Rayleigh and Euler theories. Jang and Lee [10] later extended this strategy to a disk spindle. The bases were designed with three-dimensional components so as to guarantee compliance with the beam components. Genta et al. [11] defined the dynamics of the spinning blade disks on the flexible disks. Transition elements were used in order to model structural elements such as shaft, blades and disks. Combescure and Lazarus [12] utilized a combination of 2D Fourier formulation with 3D FE elements for analyzing big rotating machine components. For asymmetric structures, 3D finite element modeling is often employed to capture a true picture of their kinematic behavior. Despite such advancements in the computing strength, a large number of sophisticated 3D models still face complicated computational issues [13].

The simplified approach of the one- and two-dimensional elements can be used in many cases, but the use of three-dimensional elements is the best method to achieve the goals of a dynamic analysis [14]. One of the advantages of using three-dimensional elements is to directly develop a finite element model of the rotating component rather than using any other mathematical approximation. The 3D elements can also address the non-linearities introduced in the rotor system of asymmetric rotors [15], and can also ensure the accuracy of the structure [16]. The asymmetric rotor can be because of the blade-off condition due to bird strike, or it can also be due to the loss of an entire blade. Another advantage of using a 3D element is that the gyroscopic effects of a rotating system can also be studied. Therefore, the use of a 3D element is the most viable solution to determine the critical frequencies of a rotating component [17].

*2.3. Bird Modeling*

2.3.1. Euler Virtual Body Formulation

In this formulation, when deformation occurs, a body flows through the reference mesh in the background, while the mesh nodes are fixed. After the deformation, the relative position of the material points and nodes changes. As the mesh nodes are fixed, mesh distortion is not possible. This technique is useful for fluids [18]. By default, the Langrangian reference frame is used in the structural simulations in ANSYS, but, in some cases where materials undergo large deformations, elements can become so distorted that a simulation cannot proceed further without using the numerical erosion of such highly distorted elements. In such cases, the Eulerian frame of reference is used. In this approach, the element grid remains stationery and materials flow though the mesh, and the mesh itself does not face the distortion problem. In the use of the Eulerian technique, a background grid is automatically generated in order to enclose all of the bodies in the simulation [19]. However, the computational time is high for this formulation compared with other techniques, such as the Langrangian formulation and the arbitrary Langrangian formulation [18,19].

2.3.2. Smoothed Particle Hydrodynamics (SPH) Technique

The Lagrangian methods track the material history by the identification of particles as a material point, and the movement of the particles is modelled by the deformation of that material. One such method to model the motion of a continuum is a smoothed particle hydrodynamics (SPH) method. The SPH model is capable of large displacements, strong discontinuities. and complex interface geometries [20]. The SPH method is based on the idea of using mesh-less particles with individual material properties, and moving as per the governing equations [21].

The SPH is implemented using a normalized corrected Kernel interpolation [22]. In order to define the SPH particles, ANSYS Autodyn is used. This allows for the user to fill the predefined bird geometry with SPH. This also allows the user to define the particle density. However, increasing the particle density may increase the simulation time.

The bird strike event is simulated by using the Eulerian formulation and SPH technique in order to compare the results of both techniques.

### 2.3.3. Bird Geometry

To consider the relationship between the bird mass and density, and bird mass and bird body diameter, the equations are as follows [23]:

$$\rho = -0.063 \ x \ log_{10}m + 1.148 \tag{1}$$

$$log_{10}D = 0.335 \ x \ log_{10}m + 0.900. \tag{2}$$

Here, $m$ denotes the mass of the bird, $\rho$ is the density, and $D$ is the diameter of the bird body.

Various authors have used one of three different artificial shapes for birds as shown in Figure 1. The International Bird Strike Research Group used an ellipsoidal [22] shape. Langrand et al. [24], McCarthy et al. [25], and Airoldi and Cacchione [26] used a hemispherical-ended cylinder, and Stoll et al. [27] considered the shape of a flat-ended cylinder.

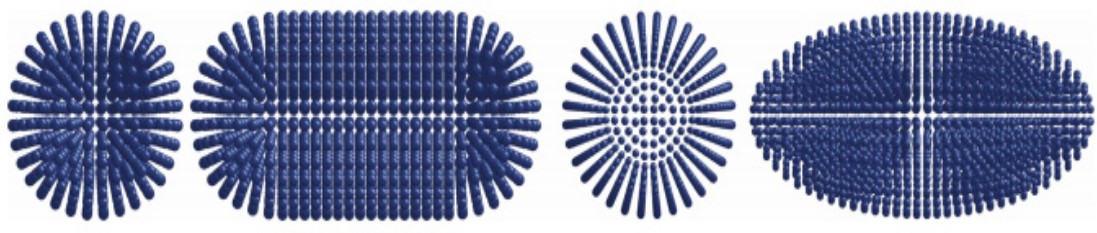

**Figure 1.** Hemispherical and elliptical bird geometry [22].

A hemispherical-ended cylinder shape was considered for the bird modelling. A bird with a mass of 0.68 kg and length to diameter ratio of 2 is used for bird strike analysis. The density of the bird is taken as 970 kg/m$^3$. For the hemispherical shape, the bird diameter was 0.081 m, and the length for the hemispherical shape was 0.162 m [22].

### 2.3.4. Probability of Bird Impacting on Different Areas of Blade

Ma Li et al. [28] categorized the blades into the following three areas: blade root, blade apex, and blade body. The probability of impact is defined by following equation:

$$P_{1a} = \frac{S_a}{S_r}, \ P_{1b} = \frac{S_{ab}}{S_r}, \ P_{1r} = \frac{S_r}{S_r} \tag{3}$$

where $P_{1a}$ is the probability of the bird impacting the blade apex, $P_{1b}$ is the probability of the bird impacting the blade body, and $P_{1r}$ is the bird impacting the blade root. $S_a$, $S_b$, $S_r$, and $S_{ab}$ are the area of the blade apex, body, root, and the total area, respectively.

Considering all of the available probabilities, the bird is most likely to impact the apex of the blade. Therefore, the bird strike on the apex of the blade is considered in the analysis [23].

### 2.3.5. Bird Material Model

Appropriate material model for the bird strike is necessary to simulate bird strike event. Bird can be considered as fluid volume for the high impact analysis. For improving the results of slicing of bird as result of strike on blades of fan rotor Isotropic-Elastic-Plastic Hydrodynamic (IEPH) model of the material should be used [22]. This material model is also used by Mc Carthy et al. [25] in their research.

Due to change in momentum of the bird upon strike on blade the forces are acted on the blades of fan rotor. For accurate modeling of hydrodynamic response of bird EOS (Equation of state) should be used [22], Murnaghan EOS is used for this purpose.

$$P = P_0 + B\left(\left(\frac{\rho}{\rho_0}\right)^{\gamma} - 1\right) \tag{4}$$

where $P_0 = 0$ is the reference pressure, $B$ and $\gamma$ are material parameters which have to be determined experimentally. The values for $B = 128$ MPa and $\gamma = 7.98$ were taken from McCarthy [25].

The material properties used for the isotropic–elastic–plastic–hydrodynamic (IEPH) model of the bird are given in Table 1:

**Table 1.** Bird material properties [22].

| Parameter | Value |
|---|---|
| Density | $\rho = 9.7 \times 10^2$ kg/m$^3$ |
| Shear modulus | G = 2.07 GPa |
| Yield stress | σ0 = 0.02 MPa |
| Plastic modulus | Eh = 0.001 MPa |

### 2.4. Engine Rotor Modeling and Material Properties

The engine used in the analysis consisted of one stage of turbo fan, five stages of compressor assembly, and two stages of turbines. The 3D CAD model of the engine with the materials is shown in Figure 2. The CAD model was used from GRAB CAD, and the fan was simplified to reduce the computational time [29].

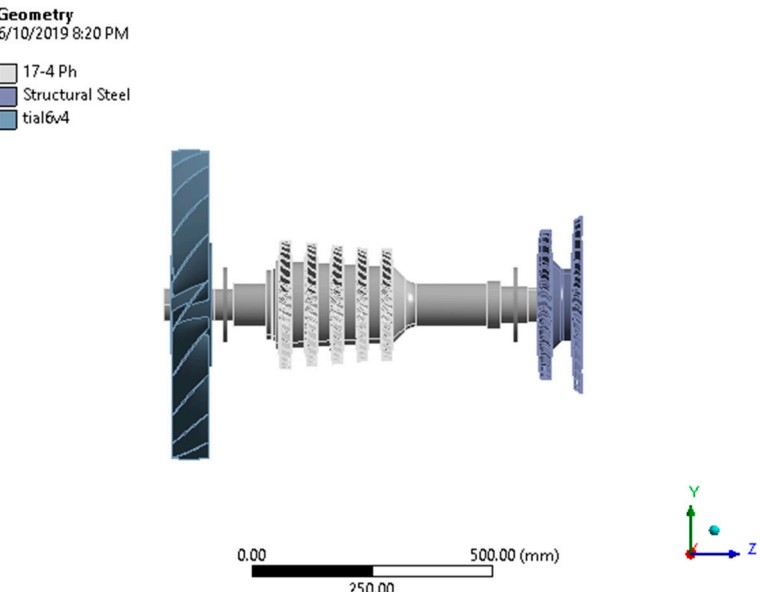

**Figure 2.** Engine CAD model and Materials.

The blades of the rotor were meshed using QUAD elements of the SOLID 185 element type, whereas the shaft was modeled using the TETRA element of the SOLID 185 element type. Several analyses were performed in order to avoid any result discrepancies due to mesh sensitivity. In these analyses, the elements through the thickness and along the length of the blades were increased so as to notice any improvement in the results. However, no further improvement was noticed, but an increase in the computational time was significant. Therefore, the total elements in the FE (Finite

Element) model were 62,480 and the total number of nodes were 25,966, with at least three elements though the thickness of the blade. The bird strike simulation was performed using the explicit dynamic solver of ANSYS. In the explicit dynamic solver, the time step is defined by the smallest element in the FEA (Finite Element Analysis) model. This solver uses the principle of energy conservation to check the solution accuracy. The overall energy of the system at each cycle was calculated, and if the energy error reached a certain threshold, the solution process terminated. By default, the threshold level was 10% of the reference energy [19]. In the simulations presented, the energy error did not reach its threshold value. The energy error may occur due to the tensile instability in the elements, as a result of the interactions of the Kernel function and the constitutive relation of the FEA model [22]. No such tensile instability was noticed within the response time of the simulation. This response is also compatible with the observations of other researchers working using SPH techniques [30,31].

In the case of the bird strike analysis, a refined mesh independent FE model was generated to obtain accurate results. The bird was modeled using SOLID 185 quad mesh, but it was taken as a Eulerian virtual body. The FE model of the bird strike analysis is shown in Figure 3.

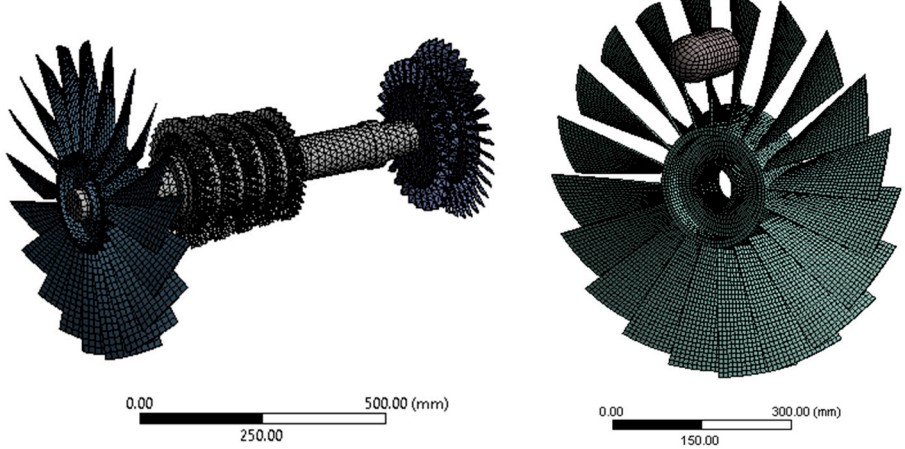

**Figure 3.** Finite element model of the bird strike.

Fan Blade Material Model

The bird strike event is as follows:

- High elastic and in elastic strains
- High strain rates
- Short duration and high impact
- Response of the structure and its interaction with high impact loads.

For a better modeling of this event, the Johnson Cook viscoplasticity model of the material was used. This model of the material is capable of defining high plastic strains and strain rates. The blades of the fan rotor were assumed to be of a Titanium alloy (i.e., Ti–6AL4V). For modeling the Johnson Cook model of the material, a Grunesien equation of state (EOS) was used [22].

$$P = \frac{\rho_o C^2 \mu \left[1 + \left(1 - \frac{\varrho}{2}\right)\mu - \frac{a}{2}\mu^2\right]}{\left[1 - (S_1 - 1) - S_2 \frac{\mu^2}{\mu+1} - S_3 \frac{\mu^3}{\mu+1}\right]} + (_o + a\mu)E \tag{5}$$

where $C$ is the velocity intercept, $S_1$ is the first slope intercept, and $\gamma_o$ is the Grunesien coefficient. The parameters that are to be used for modeling the Titanium alloy material EOS are given in Tables 2 and 3.

**Table 2.** Johnson Cook material properties (Ti–6AL–4V) [22].

| Parameter | Value |
| --- | --- |
| Density | $\rho = 4.42 \times 10^3$ kg/m$^3$ |
| Yield stress | $\sigma_y = 1098$ MPa |
| Shear modulus | G = 42 GPa |
| Strain hardening modulus | B = 1092 MPa |
| Strain rate dependence coefficient | C = 0.014 |
| Temperature dependence coefficient | m = 1.1 |
| Strain hardening exponent | n = 0.93 |
| Melting temperature | T = 1878 K |
| Heat capacity | C = 580 J/kg K |

**Table 3.** Grüneisen equation of state (EOS) parameters for Ti–6Al–4V [22].

| Parameter | Value |
| --- | --- |
| Velocity curve intercept | $C = 5.13 \times 10^3$ m/s |
| First slope coefficient | $S_1 = 1.028$ |
| Grunesien voefficient | $\gamma_o = 1.23$ |
| First order volume correction coefficient | b = 0.17 |

## 3. Simulation

### 3.1. Bird Strike Analysis

In order to see the effect of the bird strike on the critical frequency behavior of the engine rotor, a bird strike analysis was first performed. In the bird strike analysis, the fan of the engine was only considered to reduce the solution time. The bird was modeled with the Eulerian domain as a virtual body. Where the rotating velocity of the engine rotor (i.e., 18,000 RPM) was also given to the rotor. The boundary conditions of the bird strike analysis are shown in Figure 4:

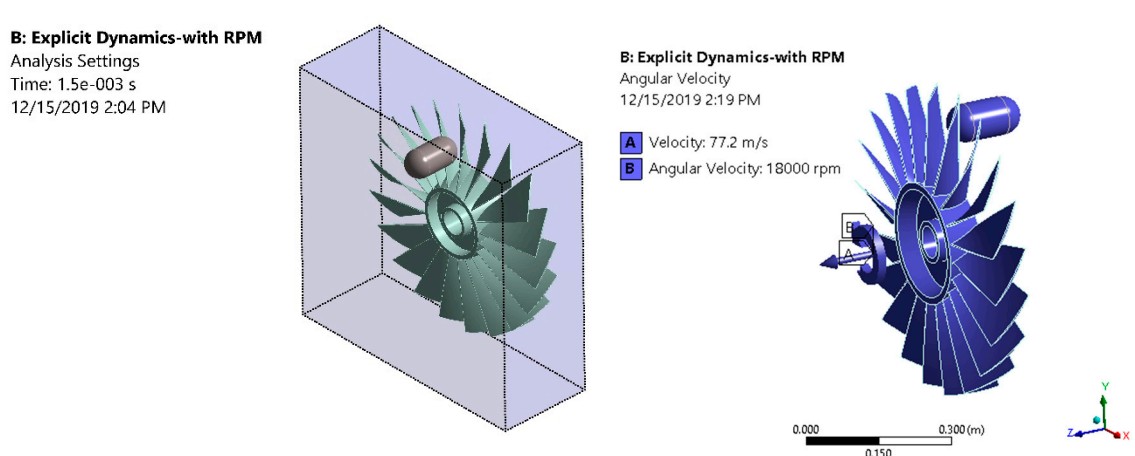

**Figure 4.** Bird strike analysis setup.

### 3.2. Modal Analysis to determine Critical Frequencies

The centrifugal forces generated in response to the rotation of the engine rotor were the main loads on the rotor. Therefore, the main load was the centrifugal forces generated against the 18,000 RPM. In the case of complex turbulent structures, Coriolis forces can also impact on the performance of the rotor, and can also add an unbalanced force into the rotor system [32]. As far as the loading conditions were considered, the load was increased stepwise in eight load steps. Gradually, the load was increased on the rotor from 0 to 21,000 RPM, with increments of 3000 RPM in each load step. As the axis of the rotor was along the global Z-axis, the load was applied only in a Z direction.

## 4. Simulation Results

### 4.1. Bird Strike Analysis

#### 4.1.1. Bird Strike Analysis Using Eulerian Virtual Body

The initial velocity of the bird was 77.2 m/s. The initial velocity was according to the take-off speed of a commercial airline aircraft [22]. The bird was aimed at 60% of the mid-span of the blades, as per the probability equations for a bird strike on an engine rotor [23]. The bird was positioned close to the fan blades to reduce the computation time. The deformation of blades is shown in Figure 5.

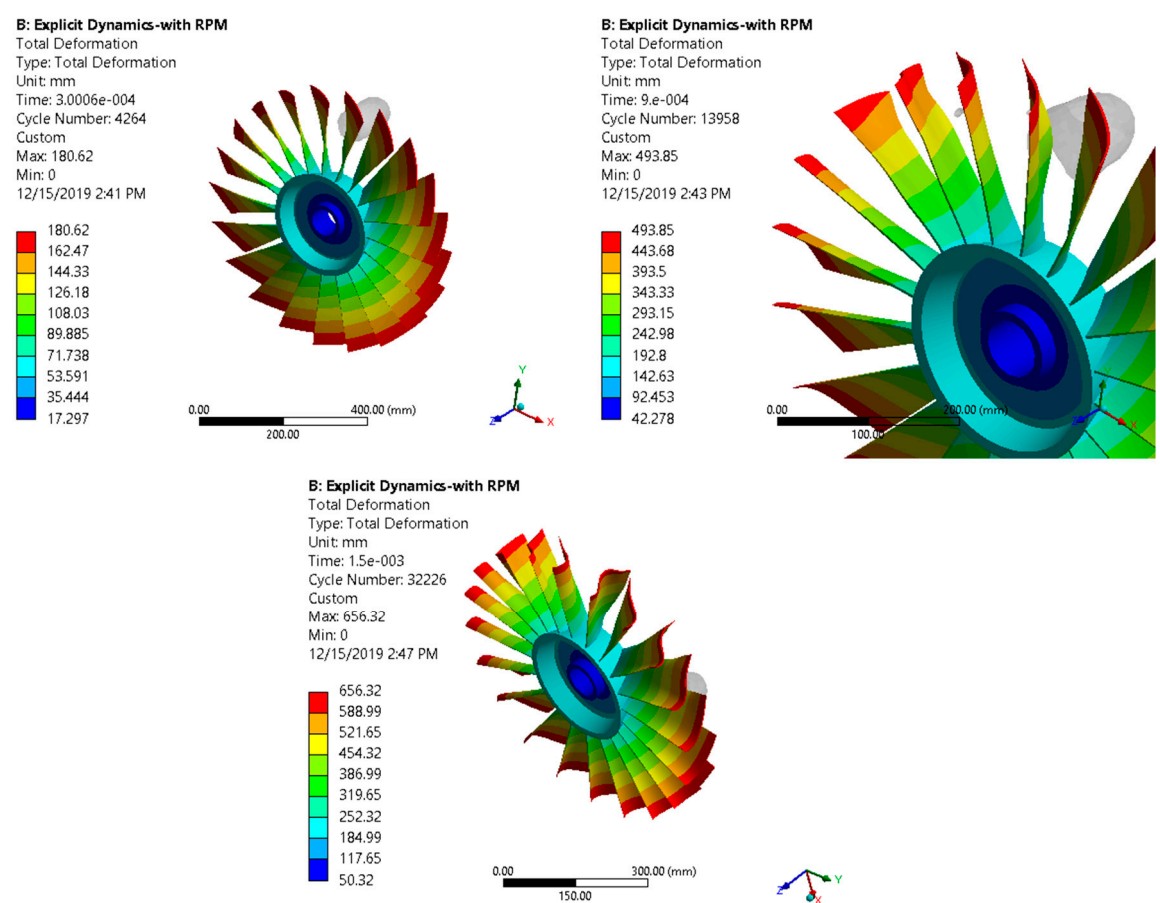

**Figure 5.** Blade deformation at 0.3, 0.9, and 1.5 m sec, respectively.

The analysis of the bird strike shows that the bird interacted with nearly seven blades. A maximum deformation was noted in the first blade, which came in contact with the bird. As the bird disintegrated afterwards, less levels of deformation were noted in the rest of the blade.

The von Misses stress generated on the blades of the fan against the bird strike are shown in Figure 6, and the von Mises stress generated in the fan blades due to the bird strike was 1255 MPa.

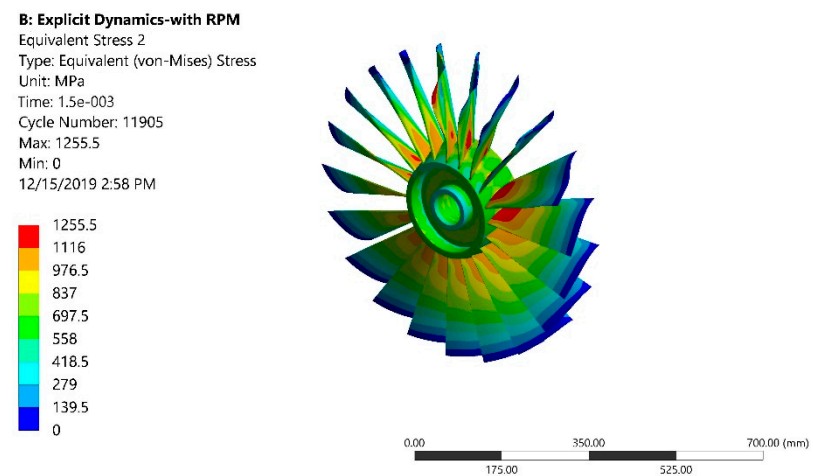

**Figure 6.** Von Misses stress (MPa).

### 4.1.2. Bird Strike Analysis Using SPH Method

The results of bird strike analysis using SPH are presented below. The deformation plots of bird strike at different time steps and disintegration of SPH are shown in Figure 7.

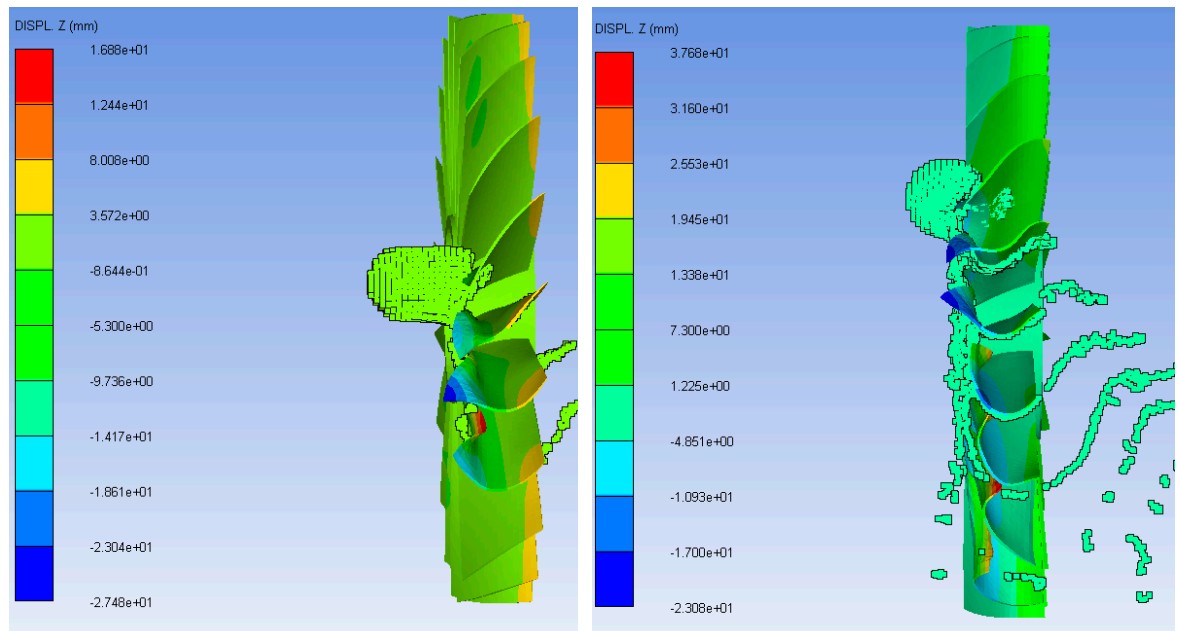

**Figure 7.** Blade Deformation in SPH model.

The deformation plots shown in Figure 7 represents the damage of the fan blades and their progression across different blades. The bird strike analysis using SPH also shows that maximum deformation or damage is observed on first blade which comes in contact with bird. In later blades the deformation or damage is reduced due to decrease in total and internal energy of fan and disintegration of bird. The loss of total energy is shown in Figure 8.

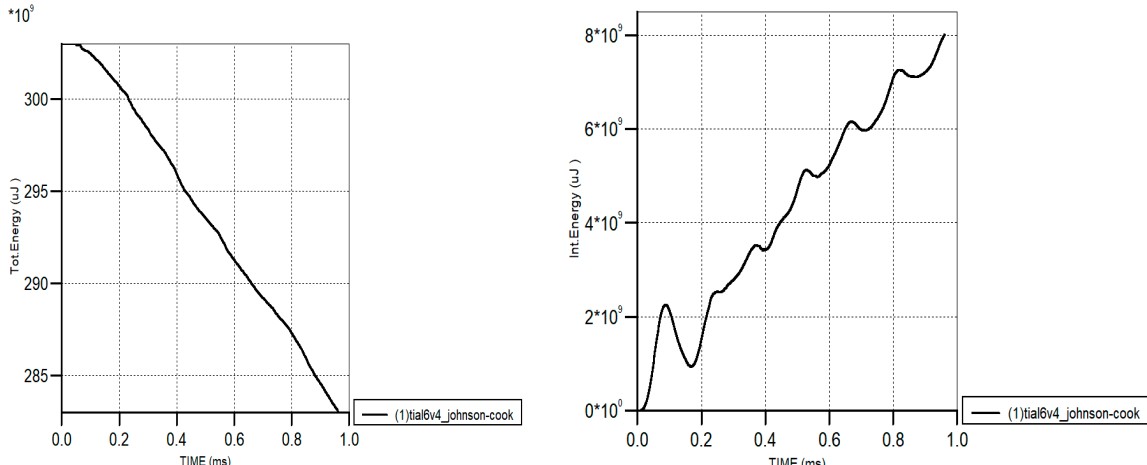

**Figure 8.** Total energy and Internal energy with time respectively.

The Figure 8 shows gradual decrease in energy of fan upon strike with bird. The initial total energy of the fan is above 300 kJ and at the end of analysis the total energy of the fan is reduced to 285 kJ. However, the internal energy of the fan is increased upon impact with bird. The increase in internal energy of the fan represents that high force is experienced by fan upon strike with bird. This may lead to high vibrations and unbalance force in the system which can affect the excitation of critical modes.

The von Misses stress generated in the fan is 1258 MPa. The von Misses stress plot is shown in Figure 9. The difference in stress values of may be due to mesh size or formulation of SPH. But the error is 0.1% which is negligible. The velocity vectors of the system are shown in Figure 10. These represents the motion of SPH particles before and after strike with fan blades.

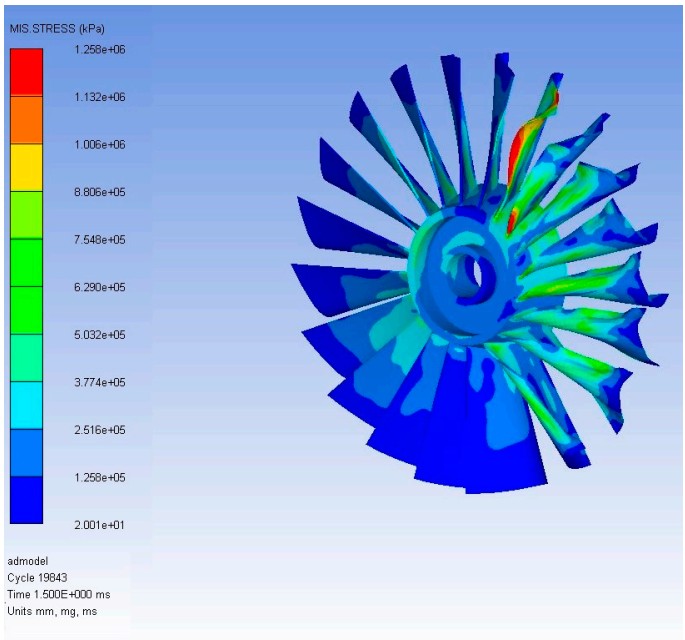

**Figure 9.** Von Misses stress (SPH).

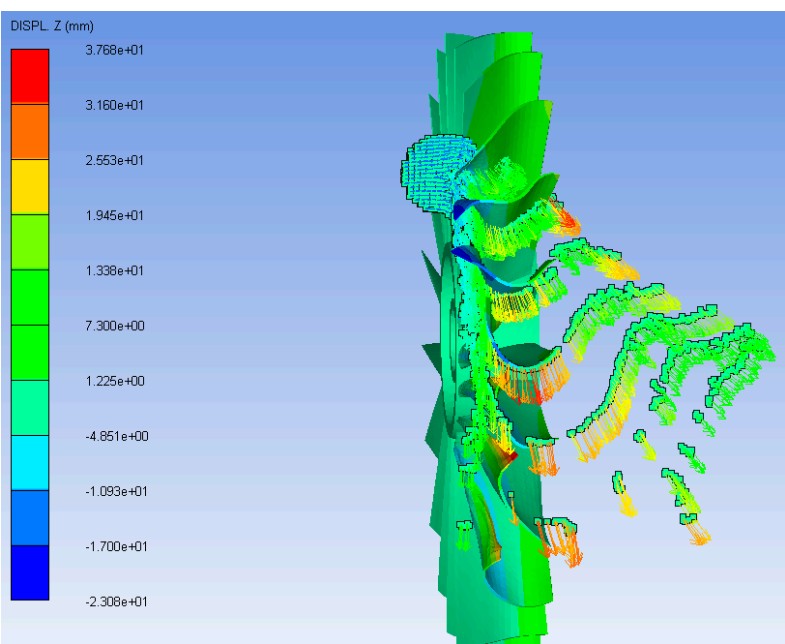

**Figure 10.** Velocity Vectors (SPH).

In order to check the reliability and validity of analyses results, the results presented for bird strike analyses are compared with the bird strike experiment. The blade was recovered from a bird strike experiment on a fan with all blades [33]. The deformed shape of the blade from experiment is compared with the simulation results. Vignjecic R. et al. [22] has adopted the same procedure for comparing their results with the experiment. They have compared their results by scanning the blade from experiment and superimposing the deformed blade on the deformation plot of the blade obtained from simulations. For checking the reliability of results presented above the deformation plots of blade effected with bird strike event are compared with deformation plots of studied carried by Vignjecic R. et al. [22]. The Figure 11 presents the plats strain plot of this study and Figure 12 presents the plastic strain plot of study carried by Vignjecic R. et al.

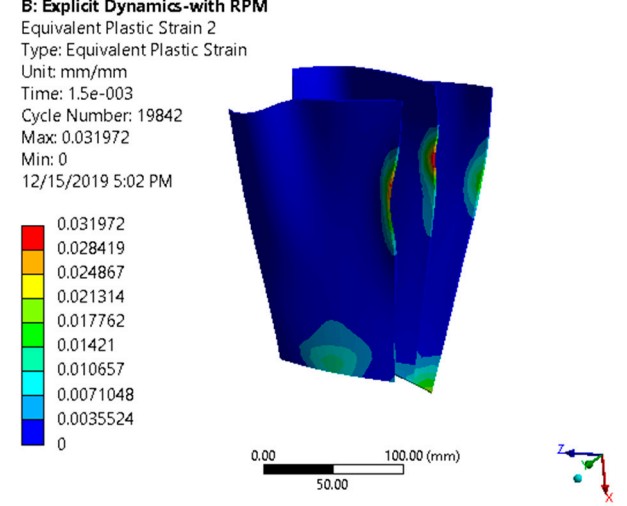

**Figure 11.** Plastic Strain plot.

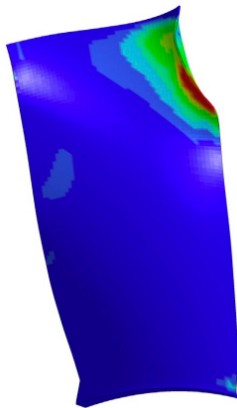

**Figure 12.** Plastic strain plot from the Vignjecic R. et al study [22].

Also, the stress plot of this study is compared with stress plots from study carried out by Vignjecic et al. [22]. The von Misses stress in this study is nearly 1255 MPa whereas the stress levels in study carried out by Vignjecic et al. is 1098 MPa. This may be due to difference is rotating RPMs. In this study the rotational velocity for engine rotor is 1885 rad/s whereas, in Vignjecic et al., has taken 806 rad/s as rotational velocity in their study [22]. Figures 6 and 9 present the von Misses stress plots of this study, whereas, the von Misses Stress plots from the Vignjecic et al. study are presented in Figure 13. The superimposition of the deformed blade from the experiment on a deformed blade from the simulations by Vignjecic et al. is presented in Figure 14.

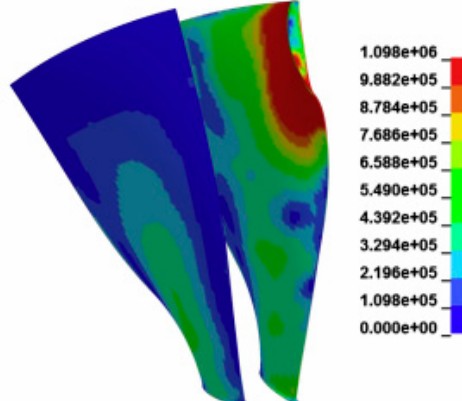

**Figure 13.** Von Misses stress plots [22].

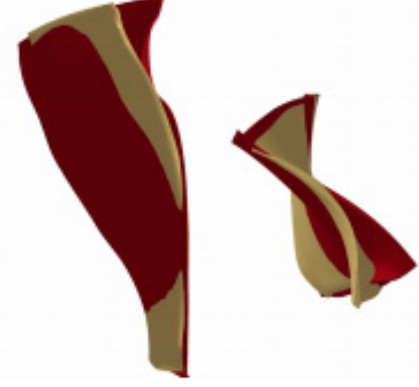

**Figure 14.** Superimposed blades [22].

The values of the stress and strain levels were different because of the difference in loading conditions, but the patterns in both studies were same, which suggests the reliability of the simulations performed in this study.

### 4.2. Modal Analysis to Determine Critical Frequencies

The CAD model of the engine rotor before the bird strike event and the deformed CAD model of the engine rotor after the bird strike are shown in Figures 15 and 16, respectively.

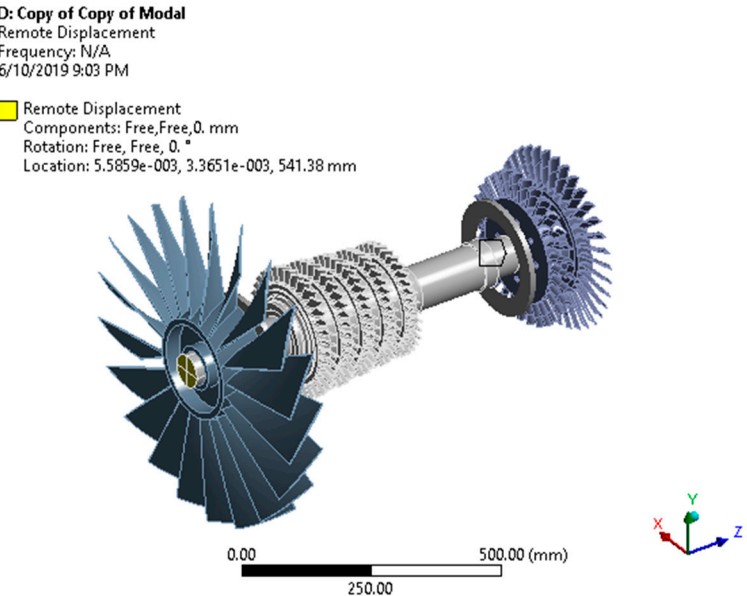

**Figure 15.** Initial model.

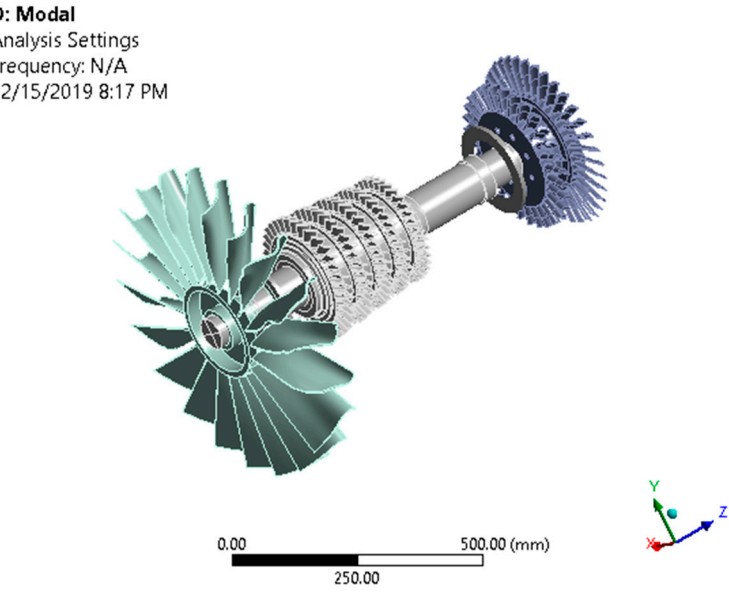

**Figure 16.** Deformed model after the bird strike.

In this case, the eigenvalue frequencies were obtained for the jet engine rotor at a rotational speed from 0 to 21,000 RPM at different load steps, using increments of 3000 RPM in each increasing load step. A comparison of the critical frequencies before and after the bird strike is shown in Table 4.

**Table 4.** Comparison of results of the modal analysis before and after the bird strike. FW—forward whirl; BW—backward whirl.

| Mode | Whirl Direction | | Mode Stability | | Critical Speed (RPM) | | Frequency @ 18,000 RPM (Hz) | |
|---|---|---|---|---|---|---|---|---|
| | Before Bird Strike | After Bird Strike | Before Bird Strike | After Bird Strike | Before Bird Strike | After Bird Strike | Before Bird Strike | After Bird Strike |
| 1 | BW | BW | STABLE | STABLE | 3058.3 | 3139.7 | 50.195 | 52.132 |
| 2 | FW | FW | STABLE | STABLE | 3074.4 | 3149.4 | 51.544 | 52.574 |
| 3 | BW | BW | STABLE | STABLE | 5401 | 6749.7 | 76.066 | 96.408 |
| 4 | FW | FW | STABLE | STABLE | 6405.7 | 7286.5 | 125.67 | 121.46 |
| 5 | FW | BW | STABLE | STABLE | 7064.7 | 7295.2 | 117.74 | 121.77 |
| 6 | FW | FW | STABLE | STABLE | 7259.4 | 7416.8 | 119.89 | 123.21 |
| 7 | BW | FW | STABLE | STABLE | 7349.8 | 7527.7 | 123.61 | 125.36 |
| 8 | FW | BW | STABLE | STABLE | 7330.5 | 8165.1 | 122.00 | 136.15 |
| 9 | BW | FW | STABLE | STABLE | 7345.2 | 8844.3 | 122.59 | 167.01 |
| 10 | FW | FW | STABLE | STABLE | 7352.6 | 8248.5 | 122.51 | 137.48 |
| 11 | BW | FW | STABLE | STABLE | 7355.3 | 8517.4 | 122.62 | 141.73 |
| 12 | BW | FW | STABLE | STABLE | 7364.5 | 8659.2 | 122.72 | 144.46 |
| 13 | FW | FW | STABLE | STABLE | 7366.6 | 8749.6 | 122.79 | 146.23 |
| 14 | BW | FW | STABLE | STABLE | 7372.3 | 8829.1 | 122.83 | 147.31 |
| 15 | FW | BW | STABLE | STABLE | 7375.7 | 8850.3 | 122.97 | 147.73 |

From Table 4, we can observe that all of the extracted modes were stable. It can be observed that before the bird strike event, the first forward whirl critical mode was excited at 3074 RPM. The second forward whirl critical mode was excited at 6405 RPM. It is noted that as the load step RPMs increased, the forward whirl frequencies increased and the backward whirl frequencies decreased. The deformation plots of the first conical and first bending mode before the bird strike event are shown in Figures 17 and 18, respectively.

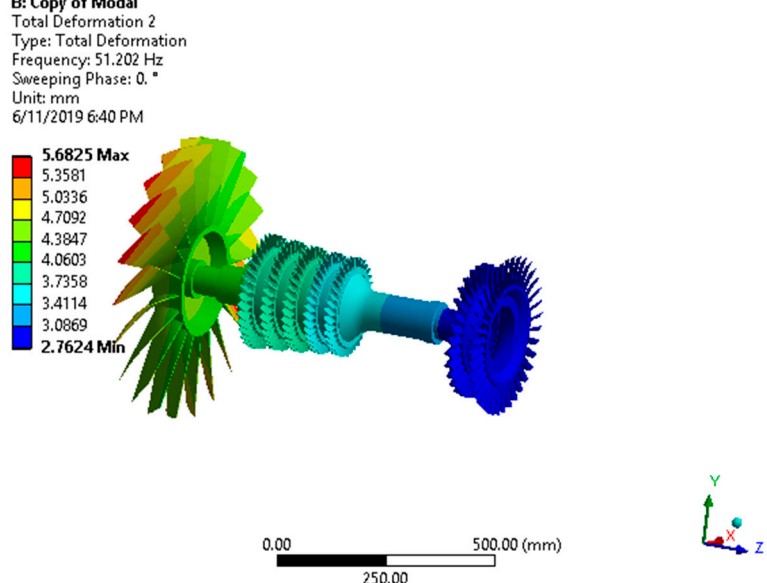

**Figure 17.** First conical mode before the bird strike.

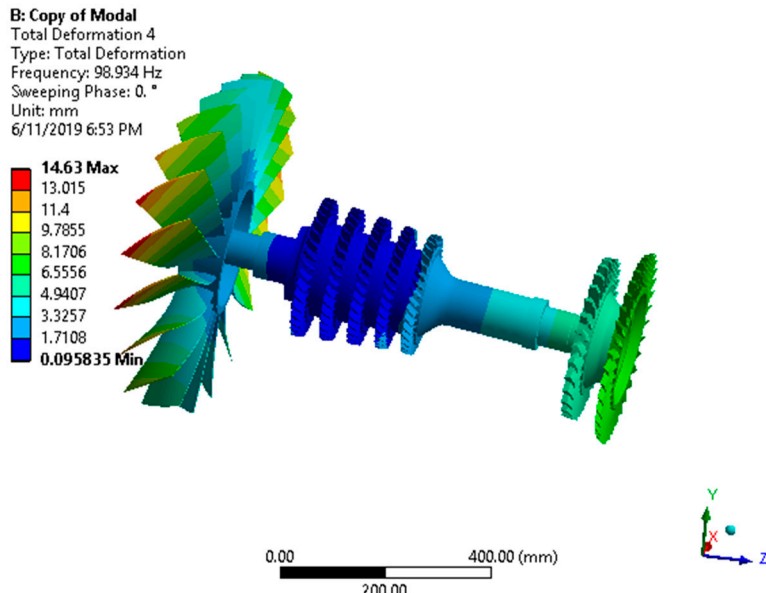

**Figure 18.** 1st bending mode before bird strike.

In the modal analysis of the engine rotor after the bird strike, the first forward whirl critical mode was excited at 3149 RPM. A difference of 91 RPM was noted in the excitation of mode. Similarly, the second forward whirl critical mode was excited at 7286 RPM. There was difference of 881 RPMs in the mode excitation after the bird strike. This is a significant increase in the required RPM for the mode excitation.

Table 4 shows that the frequencies of the critical modes of the engine rotor were increased after the bird strike. Also, the critical modes were excited at higher RPMs after the bird strike compared with the values of the excitation of the critical modes before the bird strike. Table 4 also shows that the whirl direction of some of the modes was changed after the bird strike event. It is noted that there was no change in the whirl direction of the first four modes. However, the whirl direction of modes five, seven, eight, nine, elven, twelve, fourteen, and fifteen were reversed.

The deformation plots of the first conical and first bending mode after the bird strike event are shown in Figures 19 and 20, respectively.

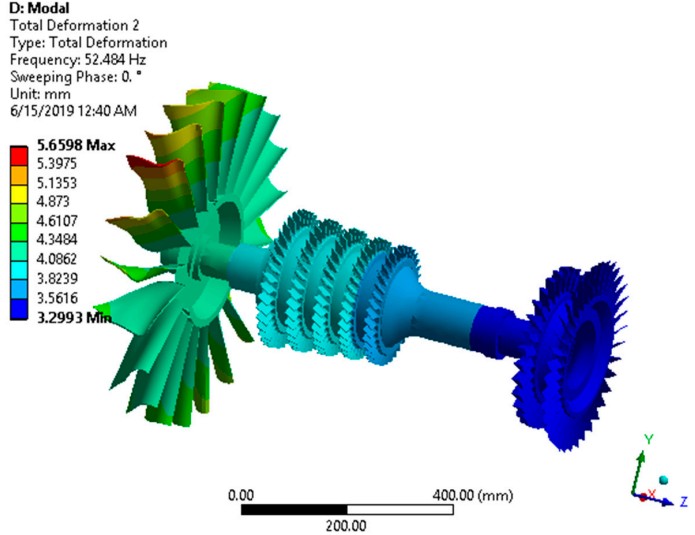

**Figure 19.** First conical mode after bird strike.

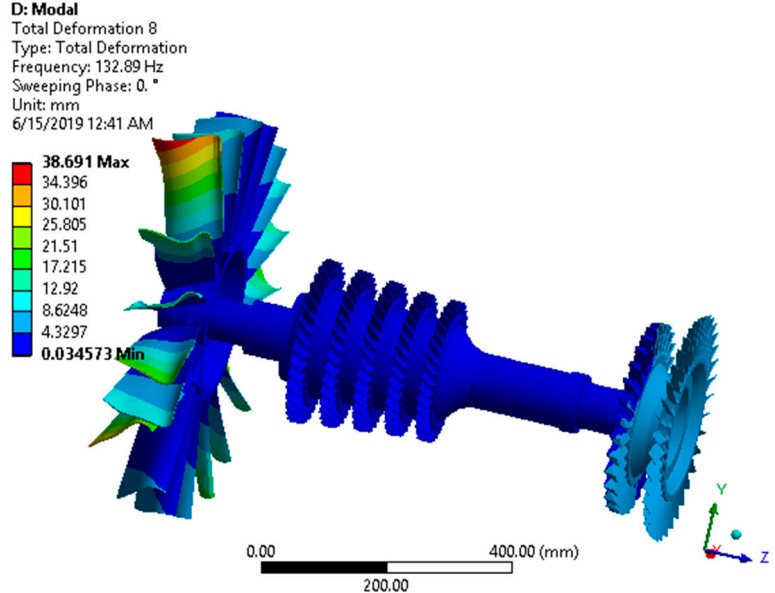

**Figure 20.** First bending mode after the bird strike.

The Campbell diagrams of the analyses before and after the bird strike are shown in Figures 21 and 22, respectively.

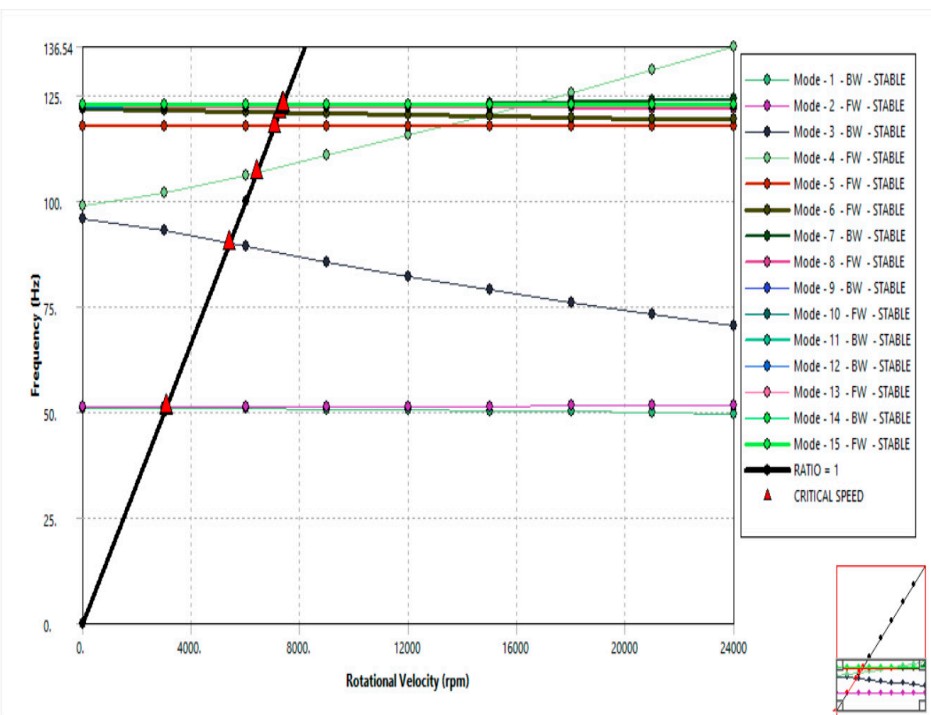

**Figure 21.** Campbell diagram before the bird strike.

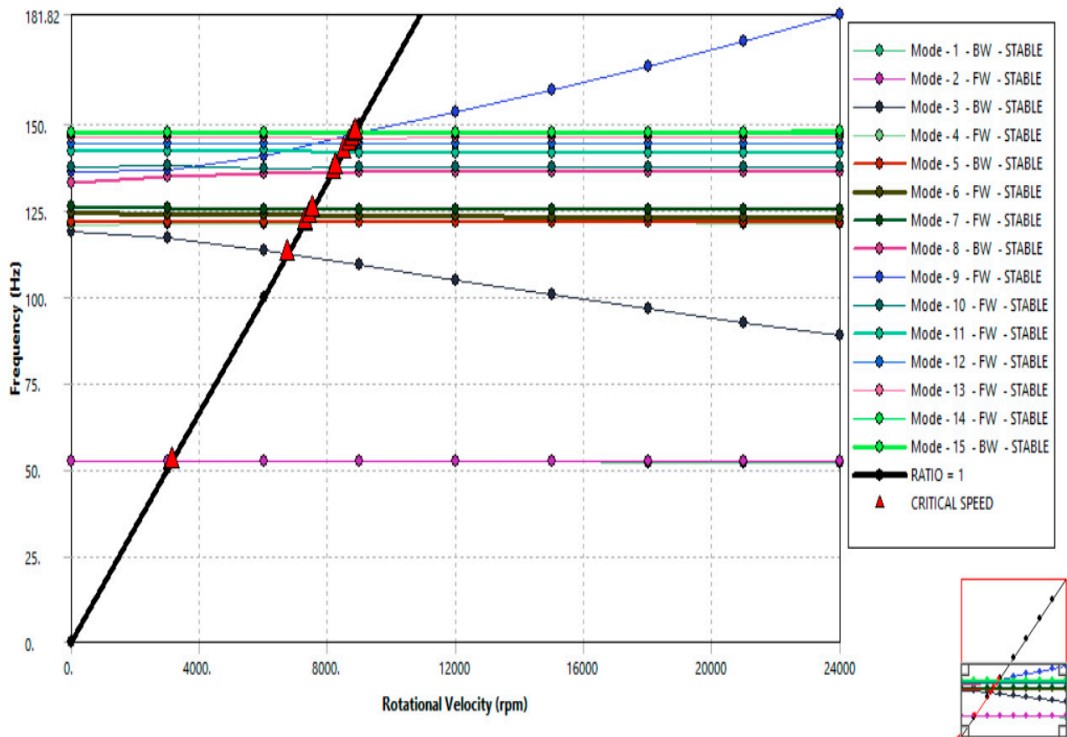

**Figure 22.** Campbell diagram after the bird strike.

## 5. Conclusions

The objective of the study was to explore the effect of a bird strike event, which may cause a blade-off condition or blade loss, on the critical frequency behavior of an engine rotor. The blade-off condition due to deformation induced as a result of a bird strike introduces unbalanced forces and non-linearities into the engine rotor system.

An aspect of the study was to model the rotor blades of the engine in ANSYS. The blade geometry impacts the dynamic behavior of the rotor significantly, therefore, it was important to capture as many geometric details of the engine blades as possible. Because of the introduction of cross section deformabilities and the predominance of centrifugal stiffening, one-dimensional models are no longer valid. Therefore, a 3D element formulation is adopted in the analyses. The best suited element type is the 3D SOLID 185 element with eight nodes. This element not only helps in capturing all of the geometric details of the engine rotor, but also helps to get accurate results at a low computational power.

In the critical frequency simulation of the engine rotor after the bird strike event, the critical modes are still conical and bending modes. However, a change in the critical frequencies of the modes is observed. An increase in the critical frequencies and excitation RPMs of each mode is observed. As the mode order increases, a greater rise in the critical frequency and excitation RPMs is observed. Also, a change in the whirl direction of the different modes is noted. A change in frequency and excitation RPM of the conical mode is not very noticeable. However, there was a significant change in the critical frequency and excitation RPM of the bending mode. Therefore, in the engine design phase, it is necessary that the engine must avoid working on a critical RPM as a result of the blade-off condition generated by the bird strike. This study only covers the blade-off condition, which means that the blade is deformed because of the bird strike making the engine rotor asymmetric; therefore, a blade loss condition may produce different results.

**Author Contributions:** Conceptualization, S.B., A.N. and A.F.R.; Formal analysis, A.N.; Funding acquisition, A.F.R.; Investigation, A.N.; Methodology, I.U.H. and S.A.M.; Project administration, S.B.; Resources, A.F.R., I.U.H. and S.A.M.; Software, A.N. and A.F.R.; Supervision, S.B.; Writing—original draft, A.N.; Writing—review & editing, S.B., A.F.R., I.U.H. and S.A.M.

**Funding:** This work and APC was funded by the Deanship of Scientific Research (DSR), King Abdulaziz University, Jeddah, under grant no. (DF-804-135-1441).

**Acknowledgments:** This project was funded by the Deanship of Scientific Research (DSR), King Abdulaziz University, Jeddah, under grant no. (DF-804-135-1441). The authors, therefore, gratefully acknowledge DSR technical and financial support.

**Conflicts of Interest:** The authors declare no conflict of interest.

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
