# Peer review of "Numerical Study on the Critical Frequency Response of Jet Engine Rotors for Blade-Off Conditions against Bird Strike"

_applsci, doi:10.3390/app9245568_

Round 1

Reviewer 1 Report

The paper presents a theoretical analysis on the rotordynamics of aircraft gas turbines at the case of a bird strike (blade-loss).

The analytical approach is well established and the methodology could render interesting results. However, it seems that the authors did not proceed to the core of results related to rotordynamic response, as the title predisposes. regarding rotordynamics, the authors present only the Campbell diagramm with the most of the results to consider mechanical integrity issues.

I expected a thorough investigation on the nonlinear response of the rotor system, with at least transient response after the bird-strike to be presented, including forces from rotor to stator, a study on the feasibility of the system to avoid a catastrophic failure.  

The authors performed a full model simulation with edge tools, and the paper must be enriched with useful results. I am expecting with great interest a thoroughly enriched version of this paper where the authors answer critical questions like: 

What are the total forces transmitted from rotor to stator during the blade-loss (bearing forces, and forces due to rubing) What are the bearing models considered in this analysis? What is the casing model? What is the time-transient response of the system, and especially o f the journal bearing during the bleade loss event?

I support the further work of the authors on the above aspects, and the resubmission of the article.

Reviewer 2 Report

The manuscript is aimed to study the numerical simulation on rotordynamic response of jet engine rotor for blade-off condition against Bird Strike. The study is interesting but some revisions are recommended as follows.

The bird strike event is analyzed using Eularian and SPH techniques. The difference of 0.1 % is noted in result of both techniques. It doesn’t mean that your study is correct according to numerical simulation without the comparison between experimental results. This is the first one and also the most important one. The description of how to apply Eularian and SPH techniques to ANSYS to simulat the model is absent. Which element type do you choose in ANSYS? How could you be sure that your simulation results converge? Equation (5) should be re-checked.

Round 2

Reviewer 1 Report

The authors did not perform a robust rotordynamic study of a jet engine after bird strike. Most of my revisions are considered for future work. I suggest that the authors address some of my comments, especially those which refer to rotordynamic response parameters (response amplitude, forces from the rotor to the bearing-casing, etc), and resubmit the article. Another alternative is to compose an article (title and content) specifically for mechanical integrity concerns without reference to rotordynamics.

Reviewer 2 Report

The manuscript is well revised according to reviewer's comments and I think it is suitable to be accepted as a journal paper.

Author Response

Thank you very much for the extended cooperation for making this manuscript ready for publication. Your keen interest and guidance have made the manuscript much more understandable and presentable than the very first version.